# Implementation of a Multicomponent Otago-Based Virtual Fall Reduction, Education, and Exercise Program (MOVing FREEly) in Older Veterans

**DOI:** 10.3390/geriatrics8060115

**Published:** 2023-11-28

**Authors:** Katherine C. Ritchey, Amanda Olney, Sunny Chen, Erica Martinez, Michelle R. Paulsen, Jennifer Tunoa, James S. Powers

**Affiliations:** 1Geriatric Research Education and Clinical Center (GRECC), VA Puget Sound Health Care System, 1660 S. Columbian Way, Seattle, WA 98108, USA; sunny.chen@va.gov (S.C.); erica.martinez@va.gov (E.M.); michelle.paulsen@va.gov (M.R.P.); jennifer.tunoa@va.gov (J.T.); 2Division of Geriatrics and Gerontology, Department of Medicine, University of Washington School of Medicine, 325 9th Ave, Seattle, WA 98104, USA; 3Rehabilitation Care Services, VA Puget Sound Health Care System, 1660 S. Columbian Way, Seattle, WA 98108, USA; amanda.olney@va.gov; 4Geriatric Research Education and Clinical Center (GRECC), VA Tennessee Valley Health Care System, 1310 24th Avenue South Nashville, Nashville, TN 37212, USA; james.powers@va.gov; 5Division of Geriatrics, Vanderbilt School of Medicine, 2147 Belcourt Ave., Suite 100, Nashville, TN 37212, USA

**Keywords:** fall prevention, telerehabilitation, veteran affairs, quality improvement

## Abstract

Purpose: The COVID-19 pandemic limited access to community fall prevention programs, thus establishing the need for virtual interventions. Herein, we describe the feasibility, effectiveness, and acceptability of a virtual, multicomponent fall prevention program (MOVing FREEly). Methods: A team of clinical falls prevention experts developed a six-week multicomponent fall prevention exercise and education class for older community-dwelling adults at risk of falling. Feasibility was measured through class attendance; effectiveness was measured through changes in performance measures, self-report of falling risk, and concern about falling; acceptability was assessed through questionnaires completed immediately upon program completion and at a three-month follow up. Results: A total of 32 patients participated in the MOVing FREEly program. Attendance for education and exercise classes on average was greater than 80% with little attrition. Patient reported reduced concern of falling, improvement in the falls efficacy scale—international (FES-I) short form, and had statistically significant improvement in 30 s sit-to-stand and single-leg balance tests. The program was well received by participants, saving them significant time and costs of travel. Conclusions: A virtual, multicomponent fall prevention program is feasible and acceptable and effective as reducing falling risk. Future studies can explore the ability of this program to reduce falling incident and injury.

## 1. Introduction

Falls are common among veteran older adults, and result in serious consequences [1]. They are the leading cause of fatal and nonfatal injury in older adults and precipitate functional decline, psychological stress, and loss of independence [2]. One in three persons over the age of 65 and one in two over the age of 80 will fall each year [3]. Falls continue to be the leading cause of injury related morbidity and mortality [3]. Recent trends also suggest that mortality from falls is highest for the oldest age groups (>75 years) and overall rates of death from falls continues to increase [4].

Falls have adverse effects on mobility, independence, and quality of life, but are largely preventable [5]. Multicomponent interventions deliver a standardized (i.e., non-individualized) combination of fall prevention interventions and are effective (e.g., exercise focused on strength and balance; home safety hazard reduction; polypharmacy reduction) [1,6,7,8]. The Centers for Disease Control (CDC) “Stopping Elderly Accidents, Deaths and Injuries” (STEADI) has endorsed several evidence-based multicomponent fall prevention programs (e.g., Stepping On; Matter of Balance) in the hope of improving community-based dissemination and engagement. The closure of many community centers during the SARS-CoV-2 pandemic limited access to in-person programs, thus highlighting the need to adapt fall prevention programs to virtual-based platforms.

Even before the pandemic, studies suggested that telerehabilitation (i.e., the delivery of rehabilitation services via telehealth modalities) is feasible and efficacious. A systematic review of nine studies which explored the concurrent validity and inter- and intra-rater reliabilities concluded that several assessments (e.g., pain, swelling, range of motion, muscle strength, balance gait, and functional assessment) were technically feasible and valid over telerehabilitation modalities [9]. A corresponding systematic review and meta-analysis of randomized physical therapy trials suggested that therapeutic interventions for physical function decline and/or disability delivered over telerehabilitation performed as well as usual care and produced similar long-term benefits [10]. Telerehabilitation is cost-effective, may improve participation in rehabilitation programs, and offers additional benefits to caregivers of persons recovering from a disabling health event such as a stroke [11,12,13].

The Veterans Health Affairs (VHA) system has been a leader in telemedicine deployment, supporting end users (i.e., veterans), infrastructure (i.e., devices, internet, and software) and staff (i.e., training and practice support and productivity/reimbursement) in a wide variety of therapeutic areas, including telerehabilitation [14]. Telemedicine is an accepted way to provide and receive care in VHA, and in many cases is a preferred modality due to transportation and other logistical challenges that rural veterans face [15,16,17]. Though there has been some success in converting other Veterans Affairs (VA) mobility and exercise programs to virtual platforms, a virtual fall prevention group program has yet to be piloted in VHA [18,19]. The goal of this project was to evaluate the feasibility and acceptance of a novel VHA multicomponent and interdisciplinary virtual fall prevention program, developed to increase awareness of fall risk factors, improve strength and balance, and promote risk-reducing behaviors.

## 2. Materials and Methods

### 2.1. Setting and Participants

This quality improvement project was piloted at VA Puget Sound, located in Seattle, Washington. This hospital is a large tertiary Veterans Health Administration (VHA) facility, which serves a large, geographically diverse area ranging along the entire Puget Sound and Western Cascade mountain range, has an inpatient hospital, skilled nursing facility and outpatient clinics including primary care, subspecialty medicine and rehabilitative services. The Seattle facility hosts a Geriatric Research, Education and Clinical Center (GRECC) and Telerehabilitation Enterprise Wide Initiative (TREWI) hub, both of which are adept at implementing novel telemedicine programs across a large, often rural, geographical region [20].

Participants at risk of falling, as defined by one or more CDC STEADI key fall risk questions (https://www.cdc.gov/steadi/pdf/steadi-algorithm-508.pdf; accessed on 28 October 2023), were referred to the program from outpatient primary, geriatric, and rehabilitation clinics [21]. A physical therapy assistant (PTA) screened participants for the following exclusion considerations: in a wheelchair 50% of the or the day or more; requires moderate to full assistance with transfers; evidence of significant cognitive impairment or dementia through chart review or basic screen; not community-dwelling. A full description of the class was offered, and patients were asked to accept or decline participation at that time. Patients with various telemedicine barriers (e.g., lacked experience with the VA Virtual Care (VVC) platform or telemedicine visits; did not own a device (i.e., tablet; computer) with microphone and camera; or lacked Internet service provider) were offered training by the PTA and/or ordered devices with Internet through a central VA distribution center.

From January 2022 to January 2023, 32 veterans were enrolled into MOVing FREEly (a total of four class cohorts). Complete data were analyzed for 27 participants who were older (mean age 75 years), mostly male (89.0%) and white (85.2%), and were at moderate or high risk of falling based on a prior history of falls or self-report response to STEADI fall risk questions (Table 1). Most (70%) had never had a VVC visit in the past. Average score on the FES-I short form indicated a high level of concern for falling when completing ADLs (14.67, SD +/− 5.0) 32. Seven participants (35%) reported a fall while participating in the series but no injuries. There were no falls or injuries during any of the exercise sessions.

### 2.2. Program Design and Implementation

The MOVing FREEly (Multicomponent, Otago, Virtual, Fall Reduction, Education and Exercise) program is a six-week multicomponent fall prevention education and exercise class. An interdisciplinary team consisting of a geriatrician, pharmacist, occupational and physical therapist (PT), all with a training in geriatrics and fall prevention, reviewed evidence-based fall prevention interventions and programs and developed class curriculum, presentation materials and participant education and exercise handbooks [22,23,24]. The elements of a successful group-based intervention (e.g., engaging participants in an active manner which is less prescriptive and more contextual; using simple language and developing trust; promote self-monitoring of behavioral change; and progression of content by group leader(s), especially exercises) were incorporated into the class model [22] [23,25]. Other patient-related education materials were obtained from the CDC STEADI website (Patient & Caregiver Resources|STEADI—Older Adult Fall Prevention|CDC Injury Center, https://www.cdc.gov/steadi/patient.html, accessed 28 October 2023) [21,26].

### 2.3. Education Program

A weekly educational class focusing on different fall prevention topics met weekly over a virtual platform (i.e., VVC) (Figure 1), and was facilitated by occupational or physical therapist, physician, or pharmacist. Instruction included a standardized pres8entation intermixed with small-group discussion, and participant handbooks mirrored weekly content/topics, including activities for participants to re-enforce concepts learned during the educational class. Participants also identified fall-risk behaviors (“risky behaviors”) at the initial class and worked with group facilitators to reduce risk through specific behavioral modifications. These self-identified goals were reviewed and reinforced at the beginning of each class and recorded into the participant handbook. Homework related to the weekly topic provided additional opportunities to reinforce class concepts between each class.

### 2.4. Exercise Program

The exercise program was based on the CDC Otago Exercise Program (Otago_2023-Implementation-Guide-for-PT-1.pdf (unc.edu), https://www.med.unc.edu/aging/cgwep/wp-content/uploads/sites/865/2023/08/Otago_2023-Implementation-Guide-for-PT-1.pdf, accessed 28 October 2023), which focuses on lower extremity strengthening and balance [21]. A PTA or PT conducted virtual exercise sessions once a week on an individual or group basis, separate from the educational class. At weeks three and five, the physical therapist increased the numbers of repetitions or challenge (e.g., eyes closed with balance exercises) if the participant was willing and could do so safely. Participants were encouraged to perform exercises on their own two to three times a week.

### 2.5. Measurements and Analysis

Feasibility was defined as participation in the group, staff effort to sustain group and ability of the group to reduce falling risk. Weekly exercise and education class attendance as recorded in the electronic health record. Staff effort included administrative time by facilitators or program assistants to review referrals to MOVing FREEly, enroll participants, connect participants to the virtual platform and provide ongoing technical support throughout the 6-week class. This was capture in the monthly MOVing FREEly operational meeting notes and electronic health record when applicable. Reduction in falling risk was measured by change in performance and self-reported measures of falling risk. The program PT completed an initial virtual evaluation for all participants who opted into the program. Assessment incorporated subjective falling history and performance tests specific for falling (e.g., four staged balance test (FSBT) and 30 s sit-to-stand (STS)) according to the CDC’s STEADI guide (STEADI-Assessment-30Sec-508.pdf (cdc.gov) and 4-Stage_Balance_Test-print.pdf (cdc.gov); both accessed on 28 October 2023) [27,28]. Performance tests were conducted over video platform with PT providing guidance and demonstration as needed. Other performance tests are more difficult to conduct over a virtual modality and were omitted [18,29]. Pre-class, post class and three-month post program questionnaires completed independently by participants captured changes in self-report of gait, balance or lower extremity impairments (adapted from CDC STEADI “Stay Independent” assessment), concern for falling (e.g., falls efficacy scale—international (FES-I) short form and self-report concern for falling) and demographic information [30,31,32]. Responses were either “yes” or “no”. Paired-sample T-tests analyzed participant improvement in physical performance measures of falling risk (e.g., 30 s STS, ability to hold a single leg test) and FES-I for patients in which there was complete pre-/post-program data. Descriptive statistics were used to examine self-report of improvement in falling risk.

Acceptance of this telemedicine fall prevention program was derived from the post-program and three-month post program questionnaires, which asked participants about their experience of participating in a virtual class, likelihood to recommend the class to others, and steps taken to incorporate fall risk behaviors (e.g., home safety modifications and changes to “risky behaviors”). We calculated travel time and mileage saved by receiving the program at home vs. in person at the Seattle VA. This was calculated by estimating the roundtrip travel mileage from participant home address (as found in electronic medical record) to Seattle VA. This implementation study was determined a quality improvement project by VA Puget Sound IRB, and patient consent was not obtained. Participation was voluntary, participants were able to opt out of program evaluation surveys, and data were de-identified prior to analysis.

## 3. Results

### 3.1. Feasibility of Program

Weekly attendance was higher for the educational (mean % attendance 89%; SD ± 0.05) than exercise classes (mean % attendance 83%; SD ± 0.04). There was low attrition throughout the six weeks for education and exercise classes, and group exercise classes had a higher attendance than the individual sessions (Table 2). All participants were offered a “test call” prior to their initial VVC visit. “Test calls” could take between 20 and 40 min, and were conducted in the context of the PTA’s regular clinical practice. On average, 40% of participants needed extra help at the start of each class to sign-on to the VVC visit, which was provided by the PTA or medical support assistant. By the fourth class, all participants were self-sufficient in their ability to navigate the VVC platform, which was a finding consistent throughout all four cohorts. The proportion of participants needing support in each cohort declined throughout the course of the study period. Table 3 summarizes the operational experiences for the virtual multicomponent exercise and educational class.

All participants (100%) reported that the program helped to reduce their fear of falling, and three quarters (76.9%) indicated the program helped them improve strength, balance, or both, and had made home safety modifications (73.1%) (Table 4). At three months, most (>95%) continued to share that the class reduced their concern about falling and had increased confidence with falling risk reduction, and few (33%) had sustained a fall (Table 4). Though most stated they felt comfortable talking to their primary provider about medications that increase their risk of falling, less than half (44.6%) had implemented medication changes at the three-month post-program follow up (Table 4). There was statistically significant improvement in the 30 s STS (pre-program mean 7.8 reps (±4.5); post-program mean 11.2 reps (±3.2); *p*-value = 0.000), ability to hold a single-leg stance (pre-program mean 2.4 sec (±3.9); post-program mean 5.2 sec (±4.3); *p*-value = 0.000), and FES-I short form scores (pre-program mean score 14.7 (±4.95); post-program mean score 12.5 (±3.56); *p*-value = 0.016) by the end of the program.

### 3.2. Acceptability of Program

As for the acceptability of the program, all (100%) were “satisfied with the program” and would “refer the program to another veteran”. Most (89%) veterans stated they “preferred a virtual platform” as it was more convenient, reduced travel burden, and increased accessibility, and would not have participated if the program was not virtual. On average, patients saved a total of 72.4 miles and 80.2 min of round trips per person by not traveling to the Seattle VA.

## 4. Discussion

MOVing FREEly is a feasible and acceptable program for delivering group-based, evidence-based fall prevention education and exercise interventions. Participation in the program was high and sustained, and resulted in an improvement in objective and subjective measures of falling risk. A virtual platform was well accepted by participants, and preferred due to time and travel savings, which were considerable. Our findings are comparable to other group-based fall prevention programs, and add to the evidence supporting the efficacy of virtually based, group, educational programs for fall prevention [23,33,34]. Our adherence rates were similar (above 80%), with a similar improvement in performance measures of falling [29].

Telerehabilitation is a reasonable modality through which to provide fall prevention programs, but requires careful logistical planning and considerations. Observations from prior studies of virtual balance or fall prevention programs suggests that even persons who are cognitively impaired or new to virtual care are able to master the independent use of this modality and become technologically independent [29]. However, this does require considerable initial skill-based coaching and assistance by the study teams or facilitators [29]. Similarly, we found that many participants required pre-program support in learning the virtual platform and signing on to virtual classes, and standby assistance during the class if technical problems were encountered. Once mastered, our participants embraced the virtual component and preferred this to in-person options. Our observations also highlighted the growing familiarity of older adults with technology and virtual health care [35]. Though most were still “new” to the VVC platform throughout the course of our study period, fewer participants needed additional assistance as time progressed, suggesting a possible increase in technological literacy in our older adult population.

Our program is different from other telemedicine-based balance or fall prevention exercise classes as it includes an education class facilitated by health care professionals and operates as a group medical appointment. Thus, it required additional considerations, such as electronic health record documentation and coding; clinic setup, referral management and virtual appointment scheduling; and patient privacy and health emergency planning (Table 3). Because veteran populations have a higher prevalence of frailty and multiple chronic conditions, the delivery of wellness and educational programs by licensed providers can allow for adaptability to meet unique health care needs and patient preferences [36]. Thus, the unique features of the VHA system support the ability of a multidisciplinary team of licensed health care providers to deliver a virtual fall prevention exercise and education class, and support development of technological skills of our patients (e.g., turn on/off device and volume; set up email account; download documents; connect to a virtual platform).

There are several limitations to our program evaluation. The number of participants was small (N = 32). Given that the intent was to determine the feasibility and acceptability of implementation of a clinical demonstration program, the data reflect the initial evaluation of this program, and were not designed to test the effectiveness at reducing the rate of falling. Future randomized control trials are needed to determine if the incidence and rate of falling are reduced in participants vs. non-participants. Secondly, the limited diversity of our participant population and health care system may limit the generalizability to Black, Indigenous, and People of Color (BiPOC) communities, transgender persons, and cis-women. As our region diversifies and our program increases in enrolment, we are hopeful to evaluate the experiences of those with diverse backgrounds and continue to incorporate inclusive language into our education and exercise curriculums. Lastly, the evaluation of this program and operational observations shared are limited to a single VA facility with telemedicine experience and infrastructure. Future work will need to explore and establish the feasibility, efficacy, and acceptability of this program at sites throughout VHA and other non-VA health care systems.

## 5. Conclusions

The MOVing FREEly program is a feasible, virtual option for offering a multicomponent fall prevention program, which reduces measures of falling risk and was well received by veteran participants. By utilizing telemedicine, this program has the potential to expand the access to traditional fall prevention interventions and better serve an increasingly aging and frail veteran population. Future studies will need to examine if this program is feasible throughout VHA and if the improvements in performance and self-reported measure of falling risk observed in this study translate into reductions in future falls.

## Figures and Tables

**Figure 1 geriatrics-08-00115-f001:**
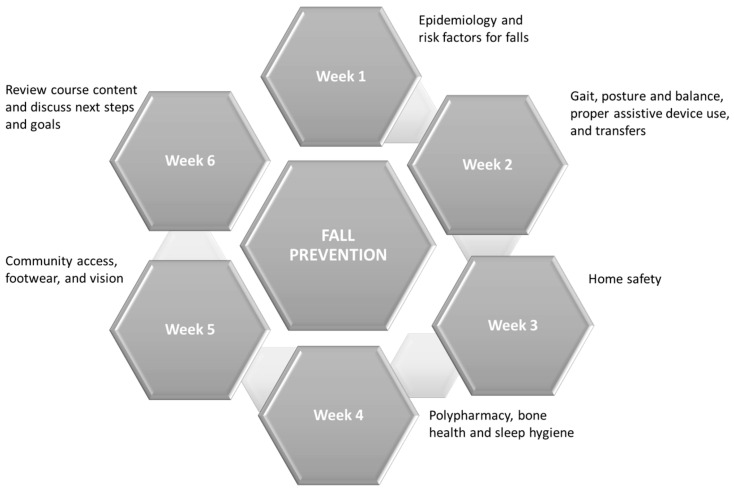
MOVing FREEly weekly class structure.

**Table 1 geriatrics-08-00115-t001:** Participant characteristics.

Sex, Male, n(%)	24 (89.0%)
Age, mean (SD, range)	75 (5.11, 64–85)
Race, white, n (%)	23 (85.2%)
Time to Seattle VA, minutes, median (SD)	80.2 min (±45.7)
Fallen within the last 12 mo, n (%)	25 (92.6%)
Fall that caused an injury, n (%)	8 (29.6%)
Fear of falling, n (%)	22 (81.5%)
Unsteady when walking, n (%)	23 (85.2%)
Use arms to stand up from a chair, n (%)	21 (77.8%)

Participant characteristics for study population for whom there was complete data (N = 27).

**Table 2 geriatrics-08-00115-t002:** Attendance (%) for exercise and education class per week.

Week	1	2	3	4	5	6
Education	91	97	81	88	91	84
Exercise—overall	77	87	87	83	80	86
Exercise—group	100	100	100	83	75	83
Exercise—individual	61	78	78	83	83	89

Weekly attendance for educational session (n = 32), group exercise (n = 12), and individual exercise (n = 18) classes by week for all study participants. Two participants were already enrolled in a community-based Otago exercise class at the time of group participation; thus, total group exercise class enrollment was 30, not 32.

**Table 3 geriatrics-08-00115-t003:** Considerations for implementing a virtual fall prevention program.

Program Component	Tips	Rationale
Logistical considerations	• Review virtual skills, virtual platform features, discuss virtual etiquette • Additional staff to support class facilitator • Send virtual appointments via email rather than text message and consider reminder calls • Knowledge of local and/or health system virtual clinic requirements, documentation and coding • Verify participant emergency contacts and location	• Builds confidence in technical skills and literacy • Allows for technical support (logging on, troubleshooting audio/visual difficulties) improves presentation timeliness, reduces delays and enhances participation • Improves class attendance • Captures workload, provider productivity and quality metrics to support program sustainability • Establish methods to engage first responders in the event of an emergency
Enrollment	• Pre-class enrolment call • Enrollment on a “rolling” basis • Mail hardcopies of participant handbooks • Pre-class physical therapy evaluations	• Clarifies class expectations; improve attendance • Reduces scheduling “bottlenecks” (accessibility) • Physical handbooks are easier to use than digital copies • Determine “best-fit” for exercise class (group vs. individual) • Build rapport with individual; learn what matters to them and guide person-centered discussions • Identify individual impairments impacting safe mobility, addressing any individual concerns (e.g., issuing assistive devices)
Management and delivery of class content	• Define virtual space as a confidential space and limit external noises and distractions • Prompt discussion during sessions • Consider presenter/facilitator communication and virtual presence • Review virtual exercise safety considerations • Group participants with similar functional levels for exercise groups and limit exercise groups depending on providers’ comfort level and participants’ safety (e.g., 4 participants/group)	• Encourages participants to share experiences openly and improves participation, respect and engagement • Builds commadore between participants and facilitates group learning • Maintain eye-contact; ensure adequate lightening; visualize face and upper torso; listen with intention; share information that is simple, concise and free of jargon; repeat questions before answering • Identify methods for balance support—assistive device, chair, counter, corner of wall; utilize a caregiver to provide support and monitor exercises • Allows adequate visualization of participant bodies while exercising to ensure safety and accuracy of movements

**Table 4 geriatrics-08-00115-t004:** Response to fall risk measures post program and at three-month follow up.

End of Program (N = 26)	n (%)	Three Month Follow Up (N = 21)	n (%)
Fall during class	19 (73.1%)	Fall since class ended	7 (33.3%)
Reduced your fear of falling	26 (100%)	Reduced your fear of falling	20 (95.2%)
Improvement in strength, balance	20 (76.9%)	Continued confidence in falling risk reduction	20 (95.2%)
I feel more comfortable talking to HCP about medications	23 (88.5%)	Made medication changes	10 (47.6%)
Plan to continue exercising	25 (96.1%)	Exercising at least weekly	14 (66.7%)
I have made changes to my home environment to reduce risk	19 (73.1%)	Has continued to reduce risky behaviors	19 (90.5%)

Table represents responses to post-program questionnaires immediately at the end of the program and at three-month follow up. Response categories were organized by common themes and represent the questions asked of participants at each time point. health care provider (HCP).

## Data Availability

The data presented in this study are available on request from the corresponding author. The data are not publicly available due to the quality improvement nature of this work.

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
