# Peer review of "Implementation of a Multicomponent Otago-Based Virtual Fall Reduction, Education, and Exercise Program (MOVing FREEly) in Older Veterans"

_geriatrics, 2023, doi:10.3390/geriatrics8060115_

Round 1

Reviewer 1 Report

Comments and Suggestions for Authors

Thank you for a very interesting manuscript. Some comments below:

1. Introduction: Explain the abbreviation VA.

2. Methods: I would recommend having a separate heading Design.

3. Methods: 2.2-2.4: There seem to be an overlap between the content in these sections, in particular 2.2 and 2.4. Please review these sections and see whether they actually can be merged into one. You may consider to use the heading Program description and implementation, with Exercise and Education as subheadings. 

4. I suggest moving the first paragraph and table 1 from the Result section to Methods. There is a section Settings and Participants, please put text and table there. 

5. Results: I would suggest to structure the Results section with headings, pointing back towards the different study aims. 

6. In the end of the Result section you present data on savings in terms of fuel etc. I don't see an explanation in the methods section that connects these data to outcome variable. Please revise for clarity. Or should this text be deleted?

7. In the methods section it says that the study was a quality development study (which I question it is), but in the discussion lines 257-273 you discuss program evaluation and you also mention that you aimed for determining proof-of-concept. I would recommend that you thoroughly consider and review which type of study design you applied and what you actually expected and could expect from the study when it comes to methodology. 

Reviewer 2 Report

Comments and Suggestions for Authors

The title suggests an evaluation of the program. However, I don't see a method to evaluate the program here. Measurements were not repeated, measurements were taken several times, but before and after therapy. Is the program reliable? I don't know. Validation needs to be done, to check the repeatability of the results. 

In my opinion, it cannot be said here that the program was evaluated. Here the effects of the program were evaluated, and I don't know if this is a good program. 

A lot of dependent and independent variables were used. What is the research, or scientific problem? 

How were performance tests specific for falling (e.g. four staged balance test (FSBT) and 30 sec sit-to-stand (STS) measured? I know the tests but who and how did they measure it? 

I tried to open the link: https://www.med.unc.edu/aging/cgwep/wp- 133

content/uploads/sites/865/2018/09/ImplementationGuideforPT.pdf. Answer: "Sorry, but the page you were trying to view does not exist".
